# Flame Stabilisation Mechanism for Under-Expanded Hydrogen Jets

**Keiji Takeno [1],*, Hikaru Kido [1], Hiroki Takeda [1], Shohei Yamamoto [2], Volodymyr Shentsov [3], Dmitriy Makarov [3] and Vladimir Molkov [3]**

[1] Research Center for Smart Energy Technology, Toyota Technological Institute, 2-12-1 Hisakata, Tenpaku-ku, Nagoya 468-8511, Japan

[2] Department of Mechanical Engineering, Osaka Electro-Communication University, 18-8 Hatsucho, Neyagawa-shi, Osaka 572-8530, Japan; syamamoto@osakac.ac.jp

[3] HySAFER Centre, Ulster University, Shore Road, Newtownabbey BT37 0QB, UK; v.shentsov@ulster.ac.uk (V.S.); dv.makarov@ulster.ac.uk (D.M.); v.molkov@ulster.ac.uk (V.M.)

* Correspondence: takeno@toyota-ti.ac.jp

**Abstract:** A hydrogen under-expanded jet released from a high-pressure vessel or equipment into the atmosphere through a 0.53 mm diameter orifice results in a sustained lifted flame for pressures above 4 MPa and flame blow-out at pressures below 3 MPa. Knowledge of whether the leaked hydrogen creates a sustained flame or is extinguished is an important issue for safety engineering. This study aims to clarify, in detail, a mechanism of flame stabilisation and blow-out depending on the spouting pressure. The model of flame stabilisation is derived using measurements and observations at the flame base location by means of high-speed schlieren images, laser diagnostics, and electrostatic probe techniques. The sustained stable flame originating from the 0.53 mm orifice is characterised by the existence of the spherical flame structures with a diameter of about 5 to 7 mm that appear one after another at the flame base and outside the streamlines of the hydrogen jet. As the spouting pressure reduces to 3.5 MPa, the sustained lifted flame becomes quasi-steady with higher fluctuations in amplitude of the flame base (lift-off height). In addition to that, flame structures are moving further from the hydrogen jet outlet, with a further decrease of spouting pressure leading to blow-out. The existence of spherical flame formations plays an important role in flame stabilisation. Based on the measurements of OH radicals using the PLIF method and ion currents, multiple flame surfaces were found to be folded in the flame structures. The hydrogen jet generates the vortex-like flow near its outer edge, creating flamelets upon ignition, ultimately forming the observed in the experiments spherical flame structures.

**Keywords:** hydrogen safety; under-expanded jet; flame stabilisation; hydrogen jet flame; electrostatic probe; turbulent flame structure; laser diagnostics; high-speed schlieren imaging

## 1. Introduction

Hydrogen has a low calorific value per unit volume, about 1/3 that of methane, so it is often transported and stored at high pressures [1]. Numerous safety studies have been conducted on the release and dispersion, ignition, and combustion of high-pressure hydrogen when it is released into the atmosphere, especially to promote the deployment of fuel cell vehicles [2,3].

For hydrogen dispersion, the spatial hydrogen concentration distribution has been studied in detail by varying the orifice diameter and stagnation pressure, e.g., refs. [4–6]. Data on the combustion of released hydrogen for different sizes of release opening and storage pressure, including flame length and width, and overpressure, when the jet is ignited, have been accumulated, analysed, and published [7–15]. Moreover, the probability of establishing a sustained jet flame when attempting to ignite with an electric spark was examined [16–18]. The results of these studies revealed that the flame length when ignited

is at most 1 m in the horizontal direction for a small orifice of 0.2 mm diameter and a stagnation pressure of 80 MPa. However, it is reported that the instantaneous concentration of 4% by volume (the lower flammable limit of hydrogen in air) extends to more than 6 m from the leak position [19]. However, the relationship between horizontal flame length and a hazard distance defined by the flammable envelope size becomes complicated when the orifice diameter is larger than 0.5 mm [19]. Thus, the question of whether the ignited hydrogen jet sustains a flame or the flame fails to hold and hydrogen disperses in the surroundings with the potential to be ignited and deflagrated later is of importance for the development of prevention and mitigation strategies, as well as for underpinning regulations, codes, and standards (RCS).

To clarify the conditions for sustained lifted flame existence and flame blow-out, several studies have been conducted, e.g., [20,21]. Figure 1 demonstrates that for any nozzle diameters below 1 mm, there are two pressure limits of flame stabilisation (below the lower curve and above the upper curve), which are functions of nozzle diameter. The sustained flame lower pressure limit indicated with dotted line in Figure 1 is about 0.1–0.2 MPa regardless of nozzle diameter below 1 mm. This is consistent with the choked flow condition for hydrogen. However, the upper pressure limits on the high-pressure side are not as simple to explain [22–25]. For example, for a 0.2 mm nozzle diameter, pressure above 60 MPa is required to sustain the flame.

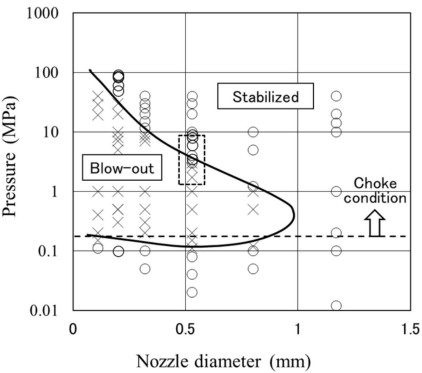

**Figure 1.** Conditions for hydrogen flame stabilisation for circular straight nozzles [20]: ○—sustained flame, ×—flame blow-out.

The criterion required for stable flames in subsonic jets was formulated as $Hw > L_{ign}$, where $L_{ign}$ is the distance from the nozzle outlet to the flame base, called lift-off height, and $Hw$ is the distance from the nozzle exit to the intersection of two lines, one being the jet axis and another being a perpendicular line to the jet axis from the maximum radial location of the stoichiometric contour of hydrogen in air in an unignited jet. This idea has been verified by experiments and simulations for subsonic jets [23]. For under-expanded jets, when the value of $Hw$ would be estimated by the notional nozzle theory proposed by Molkov et al. [26–28], the conclusions were shown to be the same [21]. The hydrogen flow rate at the upper pressure limit for the sustained flame is also reported to have approximately the same value without dependence on nozzle diameter [20]. This indicates that the notional nozzle theory of defining a virtual nozzle using physical quantities behind the shock structure is effective.

The above-mentioned criterion is useful for qualitatively explaining jet flame retention, but it does not lead to a discussion of the flame structure at the flame base, because the average concentration, which is the equivalent ratio of the hydrogen–air mixture in the jet, is used to determine the flame base. Furthermore, the flame base has a complex turbulent structure. The upper pressure limit for the sustained flame is difficult to explain, and various models have been proposed [21–25]. In the present study, therefore, the flame stabilisation mechanisms of under-expanded hydrogen jet flames are discussed by observation and analysis of the flame structures at the flame base using high-speed schlieren moving images, laser diagnostics, and electrostatic probe techniques.

## 2. Experiments and Discussion

### 2.1. Observations and Analysis of Flame Base Structures

Hydrogen was pressurized to 15∼90 MPa, stored in a 0.01 m$^3$ cylinder, and spouted to the horizontal direction from a 0.2 mm or 0.53 mm diameter nozzle installed at 1.5 m height with the pressure just before nozzle outlet regulated at a predetermined value during 10∼20 s of each experimental run. The nozzle aperture was pierced with a drill and the inside was polished by electric abrasion. The hydrogen flow rate was estimated from the pressure change of the storage cylinder before/after an experimental run.

For a nozzle diameter of 0.53 mm and a jet stagnation pressure of 8 MPa or 3.5 MPa, an axisymmetric, highly under-expanded jet flow and a shock wave structure are observed in the schlieren images shown in Figure 2. Because the pressure at the nozzle exit exceeds atmospheric pressure, the flow expands, accompanied by gas acceleration, resulting in velocities that correspond to a decrease in pressure and density. The expansion diverges peripheral streamlines from their initial direction outward from the flow centre line. A series of expansion waves are formed at the nozzle exit edge. These expansion waves are reflected as compression waves from the free surface at the jet flow boundary, which coalesce and form a barrel shock and a Mach disk. The streamlines peripheral to the Mach disk form a supersonic flow, which crosses oblique shock, and more than 90% of the total mass flow passes through this supersonic flow peripheral to the Mach disk [29,30]. In this way, a hydrogen–air mixture is formed, and the combustion at the flame base is known to be premixed, although we deal with a pure non-premixed jet of hydrogen into the air.

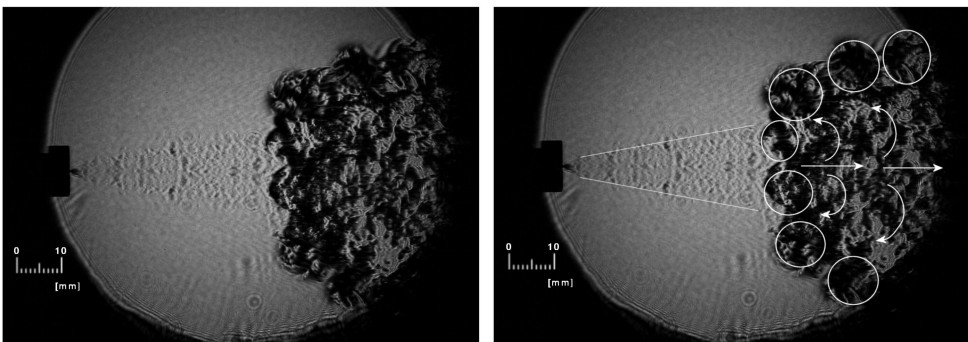

**Figure 2.** Schlieren image of hydrogen flame for steady flame condition ($d$ = 0.53 mm, $P_0$ = 8 MPa). The important phenomenon observed in the high-speed photograph are indicated with white lines and circles in the right figure.

Under steady flame conditions with a jet pressure of 8 MPa, a typical schlieren image of hydrogen flame is shown in Figure 2 (left). Figure 2 (right) demonstrates, via white lines and circles, the important phenomenon observed in the high-speed photograph. The movement of gas away from the centre of the jet in a vertical direction near the flame base is observed (see Figure 2, right). This movement is due to a combustion reaction forming spherical flame structures with a diameter of about 5–7 mm, which are indicated in Figure 2 (right) with dashed circles. Since the schlieren photograph is an integrated image through the optical path, spherical flame structures can be recognised clearly along the edge of the image. The flame structures are formed one after another in the process of combustion, and flame retention continues. These flame structures extend outward from the location of the velocity boundary of the jet (see two white lines in Figure 2, right). They cover the premixed hydrogen–air flow and are observed to be lightly rotating as they are pushed outward from the jet axis.

On the other hand, under quasi-steady conditions for flame, i.e., conditions on the upper pressure limit curve in Figure 1, with a reduced spouting pressure of 3.5 MPa, the flame structures overlying premixed air flow are not observed as stable, i.e., they appear temporarily but soon recede downstream (see Figure 3, left). The amplitude of fluctuations of the flame base position with time is much larger for quasi-steady flame conditions, i.e.,

closer to the upper pressure limit curve (see Figure 4). Figure 3 (right) shows a transient shot during the blow-out process with decreasing the hydrogen pressure down to 3.2 MPa. The flame structures extending outward from the velocity boundary are not observed, and finally, the flame is blown out, while the flame structures recede too far downstream.

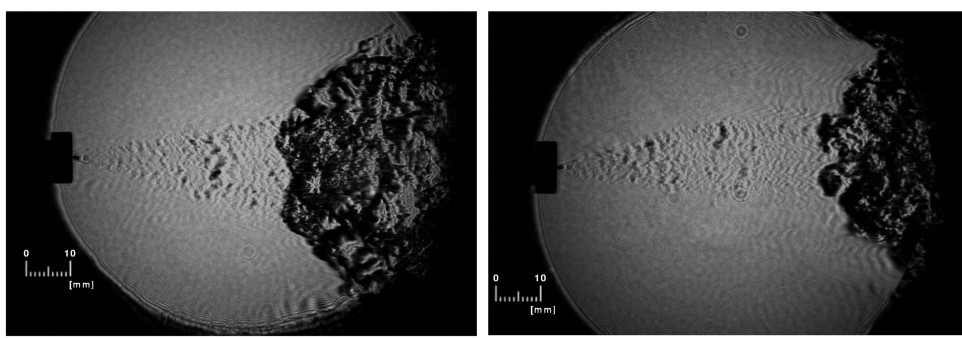

**Figure 3.** Schlieren images of a hydrogen flame for a quasi-steady flame condition (**left**, $d = 0.53$ mm, $P_0 = 3.5$ MPa) and the blow-out process (**right**, $d = 0.53$ mm, $P_0 = 3.2$ MPa).

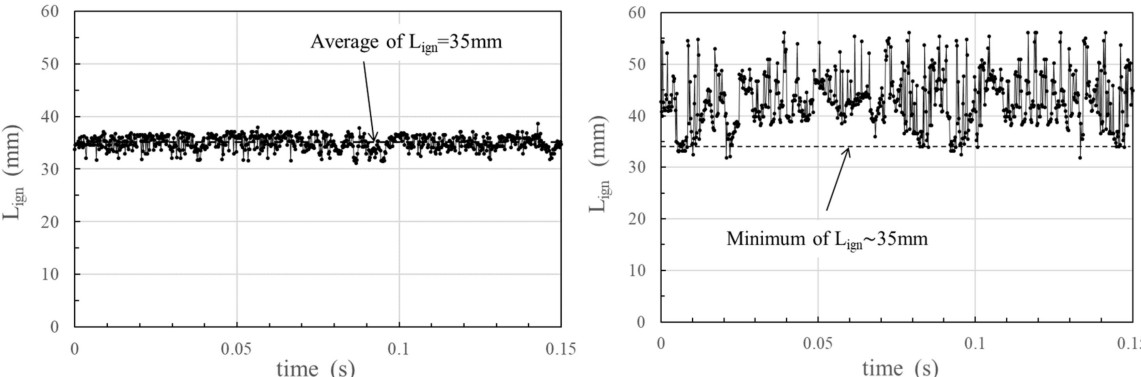

**Figure 4.** Variations of the lift-off distance with time for steady flame condition (**left**, $d = 0.53$ mm, $P_0 = 8$ MPa) and quasi-steady flame condition (**right**, $d = 0.53$ mm, $P_0 = 3.5$ MPa).

Figure 4 shows the fluctuations in time of the lift-off distance $L_{ign}$ from the nozzle outlet to the flame base derived from schlieren moving photos (20,000 f/s). The position of the flame base was determined by digitising the shading of each schlieren image. The values of $L_{ign}$ for 8 MPa are almost constant with variations of 5 mm or less, but for 3.5 Mpa, variations are as large as 20 mm. Under both conditions, the minimum values of $L_{ign}$ were found to be practically the same, i.e., 35 mm. It can be seen from Figure 4 (right) that, for 3.5 Mpa, the flame structures are oscillating with larger amplitude, i.e., moving in the nozzle direction for a moment but then immediately moving downstream to repeat it again and again. Contrary to this, at the pressure of 8 Mpa, the clump-like spherical flame structures shown in Figure 2 are formed one after another in approximately the same position shown in Figure 4 (left).

## 2.2. Analysis of Flame Base Structure by PLIF Measurement

To investigate the details of flame structure, already reported 2D cross-sectional measurements of OH radicals by Planar Laser-Induced Fluorescence (PLIF) were rearranged [12,19] and used for discussion in this study. The OH radical has a long lifetime compared to that of O or H radicals. Thus, OH has the disadvantage of identifying precise flame surface locations. However, the OH concentration is several times higher. So, the OH radicals were chosen as the target of PLIF measurement. In the experiment, the second harmonic of the Nd:YAG laser (Spectra-Physics Quanta-Ray Pro 290, maximum 1 J per pulse, 10 Hz) was converted to a shorter wavelength by using a dye laser and an SHG (BBO crystal) as the excitation source for the OH radicals. The excitation wavelength was

282.927 nm, which corresponds to the (1, 0) band absorption line Q1(6) in the $A^2\Sigma^+ \leftarrow X^2\Pi$ transition of the OH molecule, which is known to be a temperature-independent excitation line [31,32]. The 2D images of OH radicals were detected with a CCD camera (Andor Technology, Zyla, $2560 \times 2160$ pixels) with an image intensifier attached (Hamamatsu Photonics, C10880-03F). The view window is $90 \times 90$ mm and the spatial resolution of the image is approximately 0.08 mm/pixel. In these conditions, the PLIF signal becomes quasi-saturated and the approximate OH concentrations can be obtained. For more details on the use of this technique, please see previous publications [12,19].

Figure 5a shows integrated OH distribution over 20 consecutive 10 Hz laser shots under the condition of sustainable flame. The reaction zone spreads outward from the central axis of the jet with vortices. The shape and scale of vortices, and lift-off distance $L_{ign}$, could be compared to previously performed LES simulations [33–35]. Figure 5b,c show the single shot images of the region indicated by the dashed line in Figure 5a. The spherical structures of about 6–10 mm in diameter, composed of flame clusters, are observed. Although the OH distribution is the planar image cut by the central axis, it can be estimated that the spherical flame structure has multiple complex flamelets within it. In addition, the experimental conditions of Figure 5a–c are for orifice diameter of 0.2 mm and pressure of 82 MPa, (different from those in Figures 2–4), with higher under-expanded jet flow conditions, yet the fundamental phenomenon can be seen as described previously.

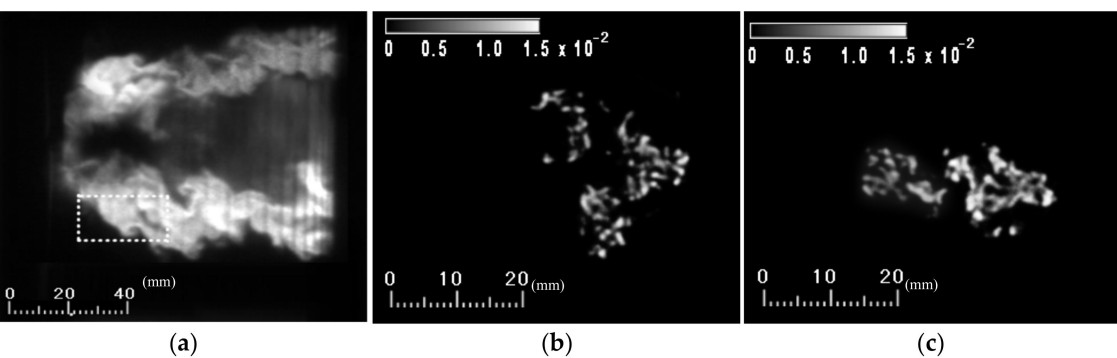

**Figure 5.** (**a**) Cross−sectional OH distribution, integrated over 20 times, of 10 Hz laser shot under the steady flame condition ($d = 0.2$ mm, $P_0 = 82$ MPa). (**b**,**c**) Cross-sectional OH instantaneous distribution of the area indicated by the dotted line in (**a**).

### 2.3. Measurement of Ion Current

The electrostatic probe technique (ion probe) is a method to detect the presence of a flame surface by capturing ion molecules in the reaction zone. The extremely high temporal resolution of this method allows us to measure the structure of turbulent flames [36,37]. Analysis of ion current waveforms obtained by the ion probe can reveal flame behaviour and turbulence structure as the flame passes through the probe. It was reported that the ion current was due to the stream of cations present in the flame [36]. It was planned to add about 1% of methane to hydrogen, but this was not necessary. The presence of trace amounts of metallic elements in the laboratory and the piping was anticipated as the likely cause, although the exact details remain unknown. The probe was inserted adjacent to the centre of spherical flame structures from below the central axis of the hydrogen jet so that it would not interfere with the jet.

Figure 6 shows the schematic illustration of the ion probe used in the experiments. The 50 μm diameter platinum wire receiving the ion signal had its tip exposed to flame only 0.5 mm from the ceramic tube. Since the ion current is weak to noise because the probe is an open circuit with high electrical impedance, it is important to improve the signal-to-noise ratio. To overcome it, a lithium-ion battery was connected in series as the power supply to detect ions in the flame at a high voltage of −50 V and the platinum wire was electrically shielded by a stainless steel pipe, 3 mm in outer diameter. From the previous results of the PIV measurements, the gas velocity measured near the flame base under the present

experimental conditions was about 20–100 m/s [30]. To obtain an accurate waveform of ion current with a spatial resolution of 0.1 mm, a data sampling velocity higher than $100/10^{-4} \times 20 = 20$ Msamples/s seems necessary if considering 20 datapoints in a cycle of signal would be necessary for accurate analysis. To accomplish it, a high-speed data recorder (HIOKI MR6000) capable of recording up to 200 M samples/s was used for data acquisition.

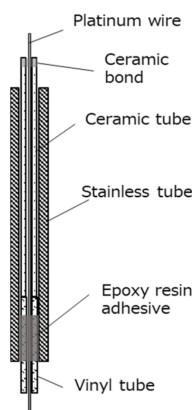

**Figure 6.** Schematic illustration of the electrostatic probe to detect the ion current in the flame.

When multiple electrostatic probes were installed, there was concern about the interference of the upstream probe to the flow around the downstream probe, but when the interval of probes was larger than 10 mm, the effects were not recognised in either the schlieren image or the ion current.

Typical cases of the relationship between the power spectral density function (*S*) of ion current fluctuation and frequency (*f*) of fluctuation are shown in Figure 7. In the calculation of *S*, $2^{16} = 65,536$ time-dependent datapoints of instantaneous ion current were analysed using the Fourier transform procedure to obtain Fourier components $X(f)$. These components were then converted to *S* with the basic equation $S = |X(f)|^2/T^2$, where *f* and *T* represent frequency of fluctuation and period of data acquisition, respectively.

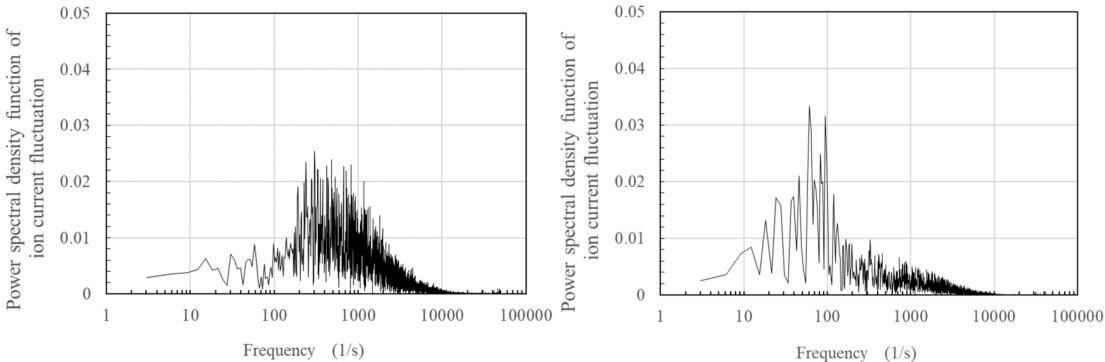

**Figure 7.** Relationship between power spectral density function of ion current fluctuation and frequency of fluctuation for steady flame conditions (**left**, *d* = 0.53 mm, $P_0$ = 8 MPa) and quasi-steady flame condition (**right**, *d* = 0.53 mm, $P_0$ = 3.5 MPa).

Under steady flame conditions with a jet pressure of 8 MPa, the value of *S* had a number of peaks from 250 to 2000 Hz. This frequency is considered to correspond to the flamelets within the spherical flame structure. When $d_m$ and $V_f$ are the diameter of flame structure and flame propagation velocity within it, the value of $V_f$ can be approximately estimated as $V_f = f \cdot d_m = (250 \sim 1500) \cdot (0.005 \sim 0.007) = 1.25 \sim 11$ m/s. This is a reasonable velocity for the turbulent flame to propagate in the premixture continuously supplied to the flame structure. However, it is not clear from only the ion current data whether the flames within a spherical flame structure are continuous or distributed.

On the other hand, under quasi-steady conditions with a jet pressure of 3.5 MPa, the values of $S$ were distributed at low values below 200 Hz. This indicates the macroscopic forward and reverse movement of the spherical flame structures in the direction of the jet, as this frequency almost agrees with that of the large movement of the leading flame base shown in Figures 2 and 3.

Cross-correlation functions were calculated to assess the similarity of the signals $x(t)$ and $y(t)$ between the two ion probes. The cross-correlation function, defined by the following equation, expresses how well two signals are correlated with a time delay of $\tau$:

$$C_{xy}(\tau) = \overline{x(t)y(t+\tau)} = \lim_{T \to \infty} \frac{1}{T} \int_{-T/2}^{T/2} x(t)y(t+\tau)dt$$

The average value of $C_{xy}(\tau)$ was set to zero during the period of data acquisition. The larger the value of the cross-correlation, the more similar signals are measured. In the present experiments, two ion probes were installed, one $x(t)$ at the centre of the spherical flame structure adjacent to the flame base, and another $y(t)$ at 5 or 15 mm downstream horizontally from the first probe. Under the stable condition with a jet pressure of 8 MPa, the cross-correlation of $x(t)$ and $y(t)$ probes with 5 mm intervals shown in Figure 8 (left) had many large peaks and the intervals of peaks were approximately 0.3 ms. If the aforementioned velocity in a flame structure is valid, the number of flamelets in a spherical flame structure can be estimated as $n = (5\sim7)/\{(1.25\sim11)\cdot0.3\} = 1\sim19$, where $5\sim7$ and $1.25\sim11$ represent the diameter of the spherical flame structures and the derived flame propagation velocity within it, respectively. This implies that several numbers of flames are folded in a swirling spherical flame structure and that some flames are independent of each other and not continuous. If all the flames within a spherical flame structure were continuous, the cross-correlation function would not have the distinct peaks like in Figure 8 (left). When the locations of two probes had a 15 mm interval, there was little cross-correlation between these two probes (see Figure 8, right). This implies that a flame structure does not correlate with the neighbouring flame structure.

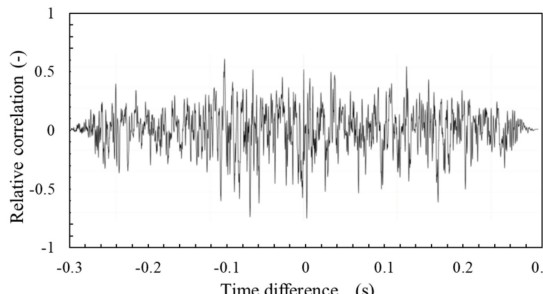 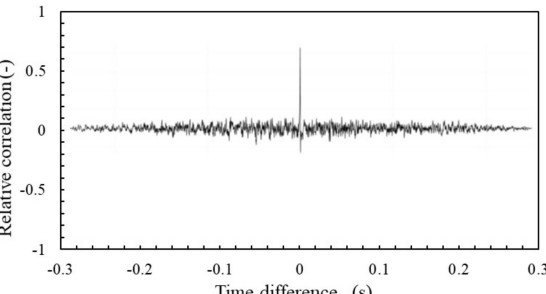

**Figure 8.** Cross−correlations of two probes with 5 mm interval (**left**) and 15 mm interval (**right**), under the steady flame conditions ($d$ = 0.53 mm, $P_0$ = 8 MPa).

## 3. Mechanisms of Flame Stabilisation

Based on the results of measurements and discussions described so far, the flame stabilisation mechanism of a high-pressure hydrogen jet flame is considered as shown in Figure 9. It highlights the dynamic and complex nature of the mixing and combustion processes used to generate the flame structures observed in the experiments. Within the air flow boundary, the high-speed air is entrained by the decelerated flow of hydrogen downstream from the Mach disk. The interaction of two gaseous flows forms a flammable premixture, which is supplied to the lifted flame to create spherical flame structures that are lightly rotating clockwise as seen in Figure 9. As a result of these complex interactions, the fast combustion reaction at the base of the flame results in the expansion of the burning structure perpendicular to the jet stream and beyond the flow boundary. A high concentration of $HO_2$ radicals, which form from the initiating reaction ($H_2 + O_2 \to HO_2 + H$) in the radical reactions of hydrogen–air, has been reported to exist in the central axis area of the

hydrogen jet flame [12,20]. The reacting premixture shown in Figure 9 receives heat from the outer spherical flame structure and the endothermic initiation reaction is considered to be promoted.

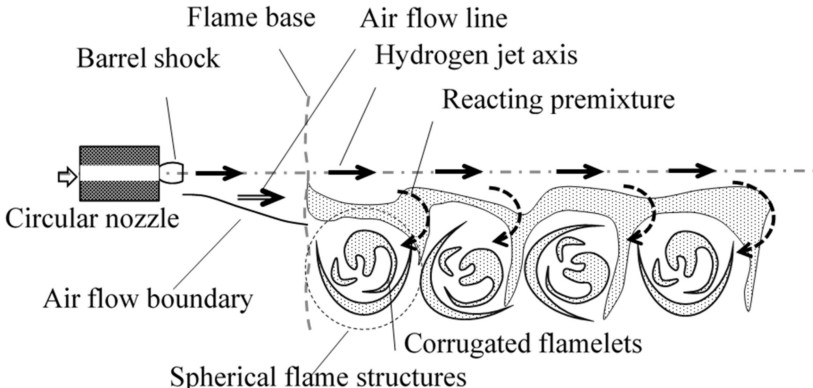

**Figure 9.** Schematic illustration of the flame stabilisation mechanism of a high-pressure hydrogen jet.

Borghi [38] used the correlation between $u'/S_L$ and $l_0/\delta_L$ to map the structure of turbulent flames, where $u'$, $S_L$, $l_0$, and $\delta_L$ represent variable velocity component, laminar burning velocity, eddy diameter, and thickness of laminar flame, respectively [38–41]. The value of $u'$ can be estimated as 10∼15 m/s from the reported LES numerical simulation [33–35]. When the other values are used from the literature, $u'/S_L$ and $l_0/\delta_L$ can be estimated at approximately 5 and 100, respectively. The Karlovitz number $K_a$, which represents the flame stretch, is defined as $K_a = g \cdot l_0/u$, where $g$ and $u$ are velocity gradient and representative velocity. When evaluated from the results of numerical simulations as well, the value of $K_a$ is approximately 0.04∼0.5. This implies that the turbulent fields do not affect the internal structure of a flame, and thus, using Borghi's diagram, it can be derived that the flamelets within a spherical flame structure are wrinkled flames with pockets, without breaking up into smaller pieces. The relationship between $K_a$ and turbulent structure has been modified by various researchers [40–44]. Peters identified the turbulent flame structure in the same region as corrugated flamelets [39,43], but the common opinion is that the flames at the flame base are corrugated without being broken up into smaller flame fragments. Even in the case of a non-reacting flow, a vortex-like flow is formed near the outer edge of the jet [33,35], and the vortices have a longitudinal vortex structure tilted in the opposite direction to the mainstream [45]. When ignited, the flamelets are generated in the vortex and they are considered to create spherical flame structures. The results of the evaluation using the diagram are consistent with the flame structure estimated by PLIF and ion probe measurements in this study.

The spherical flame structures play an important role in continuously igniting the hydrogen–air premixture to sustain the flame, as shown in Figures 2 and 3. When the hydrogen pressure was decreased, the spherical flame structures became quasi-steady with higher fluctuations in the flow direction and were swept downstream at the blow-out limit.

## 4. Conclusions

For the circular nozzle of diameter less than 1 mm, the lower pressure limit for the sustained flame is 0.1–0.2 MPa, which is consistent with the existence of choked flow at these pressures. The mechanism of flame stabilisation at the upper pressure limits being a function of nozzle diameter was investigated and discussed in this study. Based on the measurements at the flame base using high-speed schlieren images, laser diagnostics, and electrostatic probe techniques, the following results were obtained.

At the stagnation pressure of 8 MPa and nozzle diameter of 0.53 mm, the steady lifted flame was observed downstream of the shock wave structure produced by the under-expanded jet. Inside the jet boundary, the air entrained by decelerated hydrogen flow downstream the Mach disk mixed with hydrogen to form a premixture, which was

supplied to the lifted flame to create spherical flame structures with a diameter of about 5–7 mm. The clump-like flame structures were observed to appear one after another in approximately the same position. They played a crucial role in sustaining the flame.

Based on the PLIF measurements of OH radicals and ion current data, the spherical flame structures were considered to consist of complex multiple flame surfaces within it. Combining the results of previous numerical simulations with the present experimental results, the flames within spherical structures are folded and corrugated without breaking up into smaller pieces.

**Author Contributions:** Conceptualization, K.T. and V.M.; methodology, K.T., S.Y. and D.M.; experimental data acquisition, H.K., H.T. and S.Y.; numerical simulation, H.T. and V.S.; writing, K.T. and H.T.; writing check, V.S., D.M. and V.M.; supervision, K.T. All authors have read and agreed to the published version of the manuscript.

**Funding:** This research has received funding from NEDO (New Energy and Industrial Technology Development Organization). Development of Technologies for Hydrogen Production, Delivery, and Storage Systems. Next-Generation Technical Development, Feasibility Study Etc. Optimization of Regulations for FCV and Hydrogen Infrastructure, Project No. P08003.

**Institutional Review Board Statement:** Not applicable.

**Informed Consent Statement:** Not applicable.

**Data Availability Statement:** NEDO Final Report 20130000000912; Available online: https://seika.nedo.go.jp/pmg/PMG01C/PMG01CG01?startId=1667364896449&forward=1 (accessed on 20 October 2023).

**Acknowledgments:** This research was undertaken as a part of "Development for Safe Utilization and Infrastructure of Hydrogen", sponsored by NEDO (the New Energy and Industrial Technology Development Organization). Also, nac Image Technology Inc. provided tremendous support for the high-speed camera. The authors would like to extend their thanks for the sincere supports.

**Conflicts of Interest:** We the authors declare no conflict of interest, because the funder told us that we only needed to write the source of funds, etc., on the acknowledgement.

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
