# Peer review of "Flame Stabilisation Mechanism for Under-Expanded Hydrogen Jets"

_fire, doi:10.3390/fire7020048_

Round 1
Reviewer 1 Report
Comments and Suggestions for Authors
1. Figure 1, Why is the flame blown off at intermediate pressures? A more detailed explanation is needed.
2. Line 101-116, As Figure 2 doesn't show the combustion characteristics, such as "spherical flame structures with a diameter of about 5-7 mm", " lame structures are formed one after another", "flame retention" ..., mentioned in these two paragraphs, I can't understand the interpretation to Figure 2. A more detailed description of Figure 2 or more consecutive images are needed.
3. Line 162, what are the spherical structures? It is difficult to see for me.
4. Line 228, what does 5-7 stands for? What is the physical principle of this calculation?
5. The Borghi diagram should be given with the results plotted.
Comments on the Quality of English LanguageI suggest the authors make a thorough proof-reading. Some sentences need to be rephrased.
Reviewer 2 Report
Comments and Suggestions for Authors
This looks like a useful paper analyzing flame stabilization through visualization of hydrogen jet flames.
I would like to ask you to review the following points.
- There appears to be a redundant expression. Please be concise to avoid redundancy. (line 121-130)
- Please specify the units of the numbers shown in the Figure 2 and 4.
- In the description of flame stabilization(line 238), the explanation was based on the results of previous studies, but it needs to be explained in connection with the schlieren images observed in this study. Although the schlieren images are fragmentary, it is necessary to elaborate on whether the data (described in the chapter 3(line 238)) and shapes similar to the mechanism described in Figure 8 were observed.
- Check the typo errors throughout the manuscript. (line 126; Figure 4, line 147; quipped, line248; HO2)
Round 2
Reviewer 1 Report
Comments and Suggestions for Authors
Accepted